# WIDE ATTENTION IS THE WAY FORWARD FOR TRANSFORMERS?

## ABSTRACT

The Transformer is an extremely powerful and prominent deep learning architecture. In this work, we challenge the commonly held belief in deep learning that going deeper is better, and show an alternative approach that is *building wider attention Transformers*. We demonstrate that wide single layer Transformer models can typically equal or sometimes outperform deeper ones in a variety of Natural Language Processing (NLP) tasks when both are trained from scratch. The impact of changing the ***model aspect ratio*** on Transformers is studied systematically. This ratio balances the number of layers and the number of attention heads per layer, while keeping the total number of attention heads and all other hyperparameters constant. On average, across 4 NLP tasks and 10 attention types, single layer wide models perform 0.3% better than their deep counterparts. We show an in-depth evaluation and demonstrate how wide models require a far smaller memory footprint and can run faster on commodity hardware, in addition, these wider models are also more interpretable. For example, a single layer Transformer on the IMDb byte level text classification has $3.1\times$ faster inference latency on a CPU than its equally accurate deeper counterpart, and is half the size. We therefore put forward wider and shallower models as a *viable and desirable alternative* for small models on NLP tasks, and as an *important area of research* for domains beyond this.

## 1 INTRODUCTION

Since Vaswani et al. (2017), Transformer-based architectures have become widespread due to their advantages over previous architectures such as recurrent neural networks (RNNs) and sometimes even convolutional neural networks (CNNs). Many new X-formers have also been proposed that improve on the original Transformer by overcoming its limitation on sequence length by providing a more scalable attention mechanism (Choromanski et al., 2020; Wang et al., 2020b; Beltagy et al., 2020). However, little research has been done on the relevance of the size of the attention computation in each layer, the number of attention layers, and how these parameters relate to the resulting Transformer's characteristics.

The primary source of parameters in a Transformer network is the Feed-forward Network (FFN) in each encoder or decoder layer, and the linear layers which convert from the sequence feature dimension (often equal to the initial embedding dimension) to the attention feature dimension, and back again after attention is applied. Each attention head typically has an equal number of attention features. Consider an input sequence $\boldsymbol{X} \in \mathbb{R}^{S \times E}$, where $S$ is the sequence length and $E$ is the embedding dimension. Here, a multi-head attention with $H$ heads is used and each head operates on the learned projection with a dimension of $A$. After the attention mechanism there is a FFN with a single hidden dimension of size $M$. These layers are then stacked $L$ times as illustrated on the left of Figure 1. Often in a typical Transformer $E = AH$. The total number of parameters in a Transformer encoder is given by:

$$\text{Encoder Parameters} = L(3EAH + AHE + EM + ME)$$
$$= 2LE(2AH + M)$$

In this paper, we investigate the effects of changing $L$ and $H$ while keeping their product, the total number of heads, the same. We start with typical values for $L$ and $H$ and then move down to a single

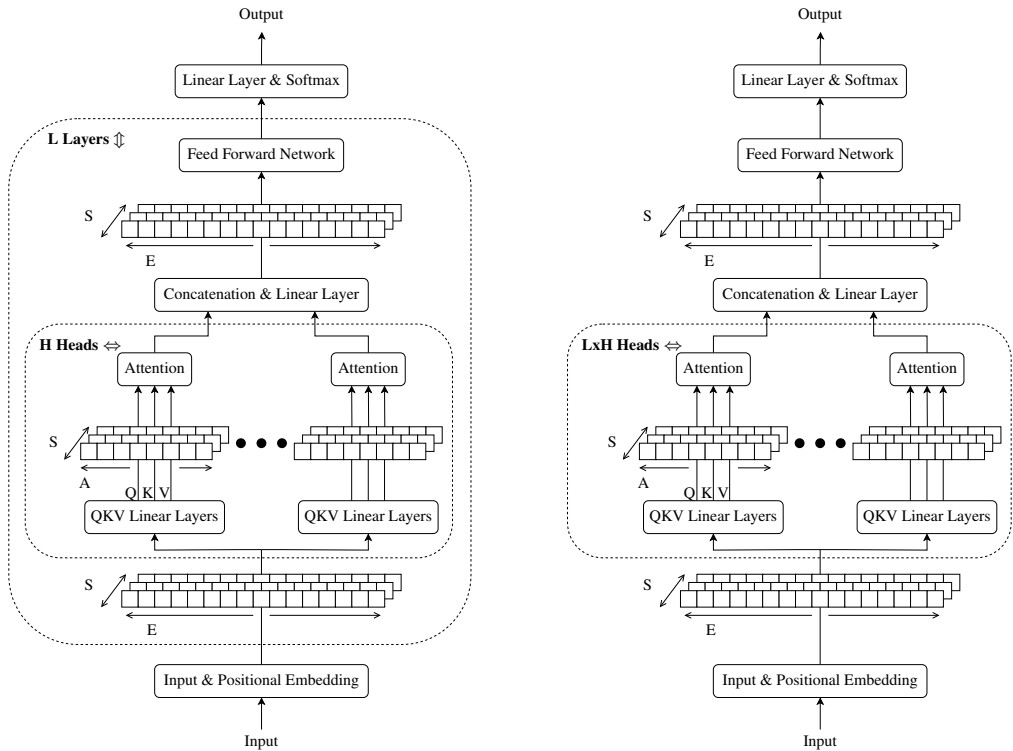

Figure 1: A comparison of a deep Transformer based classifier (left, with $L$ layers and $H$ heads for each layer) vs an equivalent wide one (right, with a single layer and $L \times H$ heads). Layer norms and residual connections have been omitted from the diagram for clarity, for details on the full Transformer architecture see Vaswani et al. (2017).

layer. A diagram illustrating our design space and the differences between our widest and deepest models is given in Figure 1. We refer to the ratio of layers to heads as the ***model aspect ratio***. This naturally leads to an intriguing question: *What is the best model aspect ratio for the growing number of X-former models?* We consider impacts of model aspect ratios on accuracy, run-time performance, model size, and interpretability.

Based on the question above, we investigate the influence of various model aspect ratios on 9 X-former models, each with their own attention mechanism, in addition to the original Transformer. Prior work on Transformer architectures has mainly focused on designing more efficient attention styles (Wang et al., 2020b; Choromanski et al., 2020) or using Network Architecture Search (NAS) to discover an optimal combination of operators (So et al., 2019). By changing the *model aspect ratio*, we consider a more coarse-grained design space. This design space is not commonly explored in the NAS algorithms for Transformers, and we evaluate some interesting model architectures such as a single layer model with many parallel heads.

For each model aspect ratio we run our experiments with each X-former across a number of text classification tasks with various input sequence lengths ranging from 500 to 4000. We empirically observe that *wider and shallower models can typically equal or sometimes beat the accuracy of deeper models*. This observation challenges the common design paradigm of trying to build deeper Neural Networks. We show several other major advantages of a shallower and wider model. First, it is more latency friendly on commodity hardware. Second, *wider models are smaller* in terms of the number of parameters. And, third, outputs of *wider models are more interpretable*.

To summarise, we make the following contributions in this paper:

- We demonstrate that wider and shallower models can typically equal or sometimes beat the accuracy of deeper models when there is no pretraining of weights or embeddings. Across

all 4 tasks, average accuracy for the vanilla Transformer increases by 0.4% between normal deep models and our single layer wide models.

- We show that our results are consistent across a variety of different attention mechanisms and input sequence lengths, and thus there is a general design equivalence in increasing the depth of a Transformer model vs increasing the width. Averaged across all non-vanilla attention types and tasks, accuracy increases by 0.3% from deepest to widest.

- We show that widening the models by fixing the attention computation size results in less overall parameters and faster inference. We show that wider models are on average $1.4\times$ smaller and have $3.1\times$ faster inference latency on a CPU and $1.9\times$ on a GPU, compared to deep models.

- We demonstrate how single layer networks can have more interpretable predictions by inspecting the attention weights of each head in a single layer.

## 2    RELATED WORK

### 2.1    WIDER NETWORKS

Zagoruyko & Komodakis (2016) and Wu et al. (2019) respectively show and investigate how widening and making shallower ResNet CNNs (He et al., 2016) can improve their performance. Transformers also use residual connections, so this helps motivate our investigation.

Xue et al. (2022b) find that wider layers in Transformers (in both attention and sequence features) can improve performance on vision and natural language tasks. However, they use a mixture-of-experts layer instead of the typical feed-forward network at the output of the Transformer encoder. They do this because a much larger dimension of sequence features means the number of parameters in the feed-forward layer would increase greatly. As we are only increasing the attention feature dimension we do not face this issue.

Xue et al. (2022a) investigate masked autoencoder training and how it can help reduce the problem of over smoothing in training deep Transformer networks - embeddings of tokens converging to be similar at deeper layers. They explore the optimal configuration of the Transformer when using masked autoencoding to pretrain it, and then fine-tune it on specific tasks. They find that for vision Transformers, it is better to pretrain with a deep autoencoder rather than a wide one. As we are not using pretrainining and pretrained embedings, their results are not directly comparable to our own.

### 2.2    ARCHITECTURE OPTIMIZATION FOR TRANSFORMERS

A lot of research in the Transformer field focused on finding a more efficient attention mechanism (Katharopoulos et al., 2020; Wang et al., 2020b; Zaheer et al., 2020; Liu et al., 2018; Beltagy et al., 2020; Choromanski et al., 2020; Kitaev et al., 2020; Tay et al., 2020a; Child et al., 2019; Tay et al., 2021), as the original dot-product attention has quadratic complexities with respect to input sequence length. Long Range Arena (LRA, Tay et al. (2020b)) compares and contrasts these methods, noting that in absence of pretrained embeddings and model parameters, the best attention mechanism is task dependent. Thus there is no clear best attention mechanism. Because of this we test different attention mechanisms on a variety of tasks with no pretrained embeddings or model parameters. This makes clear that our findings are largely independent of attention mechanism and task, and contextualise going wide as a general design decision.

The application of Neural Architecture Search (NAS) to Transformer models has also been explored. Both Neural Architecture Transformer (NAT) Guo et al. (2019) and Hardware-aware Transformers Wang et al. (2020a) use NAS to search for models that are more hardware efficient. So et al. (2019) use evolutionary search to optimize components and connections inside an encoder or decoder layer. Liu et al. (2022) apply RankNAS (Hu et al., 2021) to cosFormer (Qin et al., 2022) and standard Transformers (Vaswani et al., 2017). They perform a search of the hyperparameters on a cosFormer network and compare these to the same method applied to the standard Transformer. Tsai et al. (2020) searches the hyperparameter space of a BERT (Devlin et al., 2018) model architecture heterogeneously to find an optimal efficient network. These previous experiments have not varied or searched over different numbers of layers in the Transformer encoder, which is a key factor in what we investigate.

## 3 EXPERIMENT SETUP

### 3.1 MODEL ASPECT RATIO

For our Transformer models we fix the number of embedding features, sequence features, attention features, and the hidden layer dimension in the FFNs for each task. We vary the number of layers, $L$, and the number of heads per layer, $H$, whilst keeping $L \times H$ constant. Starting with typical values for $L$ & $H$, we then move down to a single layer with one to two intermediate model aspect ratios, observing how test accuracy changes for trained models. See Figure 1 for an illustration of our deepest and widest models.

In all of our tasks we do not use pretrained embeddings or pretrained model parameters as this allows us to make a fairer comparisons. Computing pretrained embeddings and weights that are optimised for each combination of attention and model aspect ratio would be computationally prohibitive, and using ones typically used for deep networks would introduce bias.

### 3.2 DATASTS AND MODELS

Table 1: The different tasks and datasets used.

| Task Name | Classification | Dataset | Input Type | Input Length |
|---|---|---|---|---|
| IMDb Token Level | Binary | IMDb Reviews | Review text tokens | 500 |
| IMDb Byte Level | Binary | IMDb Reviews | Review text bytes | 1000 |
| Listops | 10-way | LRA Listops | Listop bytes | 2000 |
| Document Matching | Binary | ACL Anthology | Document bytes | 4000 |

Primarily we investigate using 4 different text classification tasks, a vision based task is investigated in Section 5.5. The first two are sentiment analysis (binary classification) on the IMDb dataset. One uses input embeddings at the token level with an input sequence length of 500, and the other uses input embeddings at the byte level and an input sequence length of 1k. This second task is taken from LRA (Tay et al., 2020b), as are the final two. The third task is Listops 10-way classification with a sequence length of 2k. This task involves reasoning about sequences of hierarchical operations to determine a result, and the input is given at the byte level. The final task used is byte level document matching, a binary classification task with a sequence length of 4k. This uses the ACL anthology network for related article matching (Radev et al., 2013). We summarise each task in Table 1, further details on them can be found in Tay et al. (2020b).

For the text classification and Listops task we try four different model aspect ratios. In terms of number of layers and heads per layer these are: 6 layers, 8 heads; 3 layers, 16 heads; 2 layers, 24 heads; and finally 1 layer, 48 heads. As the matching task has an input sequence length of 4k, the models used are smaller to offset the computation size involved. Thus the combinations we use are: 4 layers, 4 heads; 2 layers, 8 heads; 1 layer, 16 heads.

In order to investigate whether the type of the attention mechanism influences the effects of widening the attention layer, we test on 10 different types of Transformer attention, including the original dot-product attention (Vaswani et al., 2017). The others are: Bigbird (Zaheer et al., 2020), Linear Transformer (Katharopoulos et al., 2020), Linformer (Wang et al., 2020b), Local attention (Liu et al., 2018), Longformer (Beltagy et al., 2020), Performer (Choromanski et al., 2020), Sinkhorn (Tay et al., 2020a), Sparse Transformer (Child et al., 2019), and Synthesizer (Tay et al., 2021). The implementations and hyper-parameter choices for each attention type are the same as used in LRA. Unlike LRA, we do not test with Reformer (Kitaev et al., 2020) due to it requiring the sequence features and attention features to have the same dimension. Training and other Transformer hyperparameters used for each task are given in Appendix A.

Table 2: Test accuracy on all tasks for deepest and widest models.

| Attention Type | IMDb Token | | IMDb Byte | | Listops | | Doc Matching | | Average | |
|---|---|---|---|---|---|---|---|---|---|---|
| | Deep | Wide | Deep | Wide | Deep | Wide | Deep | Wide | Deep | Wide |
| BigBird | 86.0 | 85.3 | 62.7 | 62.4 | 36.7 | 37.3 | 63.9 | 64.0 | 62.3 | 62.3 |
| Linear | 86.7 | 88.0 | 64.5 | 64.7 | 37.1 | 37.4 | 64.2 | 63.9 | 63.1 | 63.5 |
| Linformer | 84.2 | 84.1 | 56.3 | 52.6 | 29.9 | 37.0 | 64.5 | 63.1 | 58.7 | 59.2 |
| Local | 74.2 | 73.9 | 55.5 | 57.0 | 36.8 | 37.7 | 58.0 | 58.1 | 56.1 | 56.7 |
| Longformer | 86.0 | 84.1 | 61.6 | 57.5 | 36.8 | 37.7 | 61.2 | 58.1 | 61.4 | 59.4 |
| Performer | 87.0 | 87.0 | 64.4 | 64.8 | 36.2 | 36.6 | 65.0 | 66.3 | 63.1 | 63.7 |
| Sinkhorn | 86.1 | 86.5 | 62.3 | 61.6 | 19.4 | 21.7 | 64.0 | 65.7 | 57.9 | 58.9 |
| Sparse | 85.7 | 85.7 | 61.2 | 62.9 | 37.0 | 36.9 | 63.2 | 63.6 | 61.8 | 62.3 |
| Synthesizer | 86.7 | 86.5 | 61.4 | 61.1 | 36.5 | 37.3 | 71.1 | 72.3 | 63.9 | 64.3 |
| Transformer | 85.8 | 85.5 | 62.6 | 62.4 | 35.7 | 37.2 | 64.0 | 64.4 | **62.0** | **62.4** |
| Average | 84.8 | 84.7 | 61.2 | 60.8 | 34.2 | 35.7 | 63.9 | 64.0 | **61.0** | **61.3** |

# 4 RESULTS

## 4.1 OVERALL PERFORMANCE

Each combination of task, model aspect ratio, and attention, is ran three times with mean and standard deviations recorded. We first summarise the differences in accuracy for our deepest and widest models in Table 2.[1] For **Deep** models on the IMDb and Listops classification tasks, we mean a Transformer with 6 layers each with 8 attention heads. On the document matching task this is reduced to 4 layers each with 4 attention heads due to the 4k input sequence length greatly increasing the computational burden. These are the deepest models we consider in our setup. These model aspect ratios are also the setup used in Tay et al. (2020b). For **Wide** models, we consider a single layer network that has its number of heads matching the total number in the Deep models. For instance, if the original **Deep** model has a model aspect ratio of 6-8, its wider counterpart is 1-48. These are the widest models in our setup.

Table 2 shows that Wide models achieved +0.3% accuracy compared to Deep models, averaged across 10 different attention types and 5 different datasets. On most tasks the performance of Wide and Deep was similar, with Listops being the exception (+1.5%). This result holds for both the 'vanilla' Transformer with dot product attention, and in general for many other types of attention. Sinkhorn sees the largest average gain by going wide (+1.0%) and of all the attention mechanisms only Longformer gets worse (−2.0%).

## 4.2 PERFORMANCE BREAKDOWNS FOR EACH TASK

Performance breakdowns for each task and model aspect ratio are given in Tables 3 and 4 with standard deviations and averages.

For IMDb token level text classification, the results (Table 3, left) show that model performance is invariant to model aspect ratio, with all 4 averages across attention mechanisms being within 0.2%. Comparing the widest and deepest models for each attention mechanism, only Longformer has a >1% accuracy increase when deep (+1.9%). Conversely, only Linear Transformer has a >1% accuracy increase when wide (+1.3%).

At the byte level (Table 3, right) there's a similar picture, with all 4 averages within 0.5%. There are however some notable deviations from the pattern of aspect ratio invariance. Two models perform better by >1% in their deepest configuration than widest: Linformer (+3.7%) and Longformer (+4.1%). Likewise two models perform better by >1% when widest: Local (+1.5%) and Sparse (+1.7%).

---

[1]For standard deviations see relevant entries in Tables 3 and 4.

Table 3: Test accuracy on IMDb token level and IMDb byte level for different model aspect ratios.

| Attention | IMDb token level (layers-heads) | | | | IMDb byte level (layers-heads) | | | |
| --- | --- | --- | --- | --- | --- | --- | --- | --- |
| | 6-8 | 3-16 | 2-24 | 1-48 | 6-8 | 3-16 | 2-24 | 1-48 |
| BigBird | **86.0** ± 0.1 | 85.7 ± 0.1 | 85.8 ± 0.1 | 85.3 ± 0.1 | 62.7 ± 0.4 | 63.5 ± 0.1 | **63.8** ± 0.4 | 62.4 ± 0.2 |
| Linear | 86.7 ± 0.7 | 86.9 ± 0.3 | 87.3 ± 0.4 | **88.0** ± 0.1 | 64.5 ± 0.2 | 64.5 ± 0.2 | 64.6 ± 0.1 | **64.7** ± 0.1 |
| Linformer | **84.2** ± 0.2 | 82.4 ± 0.1 | 81.1 ± 1.0 | 84.1 ± 1.0 | **56.3** ± 3.6 | 52.8 ± 0.4 | 53.0 ± 0.0 | 52.6 ± 0.1 |
| Local | 74.2 ± 0.3 | 74.1 ± 0.1 | **74.2** ± 0.2 | 73.9 ± 0.1 | 55.5 ± 0.9 | 57.1 ± 0.1 | **57.5** ± 0.2 | 57.0 ± 0.0 |
| Longformer | **86.0** ± 0.0 | 85.8 ± 0.2 | 85.7 ± 0.1 | 84.1 ± 0.1 | 61.6 ± 0.1 | **61.8** ± 0.2 | 60.5 ± 0.3 | 57.5 ± 0.0 |
| Performer | 87.0 ± 0.1 | **87.3** ± 0.3 | 87.0 ± 0.3 | 87.0 ± 0.0 | 64.4 ± 0.1 | 64.7 ± 0.0 | 64.7 ± 0.1 | **64.8** ± 0.1 |
| Sinkhorn | 86.1 ± 0.3 | 86.4 ± 0.2 | 86.4 ± 0.1 | **86.5** ± 0.1 | **62.3** ± 0.0 | 62.0 ± 0.1 | 61.9 ± 0.2 | 61.6 ± 0.1 |
| Sparse | 85.7 ± 0.2 | 85.6 ± 0.4 | **85.8** ± 0.1 | 85.7 ± 0.2 | 61.2 ± 0.1 | 61.5 ± 0.6 | 62.1 ± 0.2 | **62.9** ± 0.2 |
| Synthesizer | 86.7 ± 0.2 | **87.2** ± 0.1 | 87.1 ± 0.2 | 86.5 ± 0.2 | **61.4** ± 0.2 | 61.2 ± 0.0 | 61.1 ± 0.1 | 61.1 ± 0.1 |
| Transformer | 85.8 ± 0.8 | 85.8 ± 0.1 | **85.9** ± 0.1 | 85.5 ± 0.1 | 62.6 ± 0.4 | **63.4** ± 0.2 | 63.3 ± 0.2 | 62.4 ± 0.6 |
| Average | **84.8** ± 0.1 | 84.7 ± 0.1 | 84.6 ± 0.1 | 84.7 ± 0.1 | 61.2 ± 0.4 | 61.2 ± 0.1 | **61.3** ± 0.1 | 60.8 ± 0.1 |

Table 4: Test accuracy on Listops and document matching for different model aspect ratios.

| Attention | Listops (layers-heads) | | | | Document matching (layers-heads) | | |
| --- | --- | --- | --- | --- | --- | --- | --- |
| | 6-8 | 3-16 | 2-24 | 1-48 | 4-4 | 2-8 | 1-16 |
| BigBird | 36.7 ± 0.2 | 37.0 ± 0.7 | 37.1 ± 0.2 | **37.3** ± 0.3 | 63.9 ± 0.5 | **64.7** ± 0.7 | 64.0 ± 1.0 |
| Linear | 37.1 ± 0.4 | **37.5** ± 0.4 | 37.3 ± 0.3 | 37.4 ± 0.5 | **64.2** ± 0.5 | 63.8 ± 0.1 | 63.9 ± 0.5 |
| Linformer | 29.9 ± 8.9 | 36.9 ± 0.3 | 36.7 ± 0.1 | **37.0** ± 0.1 | **64.5** ± 1.1 | 62.7 ± 0.4 | 63.1 ± 0.5 |
| Local | 36.8 ± 0.1 | 37.1 ± 0.1 | 37.2 ± 0.3 | **37.7** ± 0.2 | 58.0 ± 0.2 | **58.4** ± 0.2 | 58.1 ± 0.5 |
| Longformer | 36.8 ± 0.2 | 36.9 ± 0.3 | 36.9 ± 0.2 | **37.7** ± 0.2 | **61.2** ± 0.2 | 60.3 ± 0.4 | 58.1 ± 0.6 |
| Performer | 36.2 ± 0.1 | 34.1 ± 3.1 | 32.0 ± 6.4 | **36.6** ± 0.1 | 65.0 ± 0.5 | 65.1 ± 1.2 | **66.3** ± 0.4 |
| Sinkhorn | 19.4 ± 2.1 | 18.0 ± 0.2 | 17.9 ± 0.5 | **21.7** ± 3.8 | 64.0 ± 0.1 | 64.1 ± 0.5 | **65.7** ± 0.6 |
| Sparse | 37.0 ± 0.3 | 37.1 ± 0.1 | **37.6** ± 0.6 | 36.9 ± 0.0 | 63.2 ± 0.2 | **64.3** ± 0.6 | 63.6 ± 0.3 |
| Synthesizer | 36.5 ± 0.3 | 36.7 ± 0.2 | 32.9 ± 5.8 | **37.3** ± 0.1 | 71.1 ± 0.5 | 72.0 ± 1.6 | **72.3** ± 0.6 |
| Transformer | 35.7 ± 1.2 | 36.4 ± 0.3 | 36.7 ± 0.1 | **37.2** ± 0.2 | 64.0 ± 0.5 | **64.6** ± 0.3 | 64.4 ± 0.3 |
| Average | 34.2 ± 0.9 | 34.8 ± 0.3 | 34.2 ± 0.9 | **35.7** ± 0.4 | 63.9 ± 0.2 | **64.0** ± 0.2 | 64.0 ± 0.2 |

In the Listops task (Table 4, left) we see a strong tendency for wider models to do better. For this task individual models in training would often reach 17-20% accuracy and then possibly make a jump to 36-38% accuracy as training went on. Illustrated by the Linformer and Sinkhorn entries, wider models were more likely to make this jump. The data also shows us that even if all the runs of an attention and model aspect ratio combination make the jump, the wider variants still have 0.3-1.5% higher accuracy. Overall there's an average increase in accuracy of 1.5% from deepest to widest models with many models seeing a >1% increase in accuracy.

On the matching task (Table 4, right), there's a small difference in performance between wide and deep networks with the three averages being within 0.1% of each other. Two models perform better by >1% when deepest than widest: Linformer (+1.4%) and Longformer (+3.1%). Three models perform better by >1% when widest than deepest: Performer (+1.3%), Sinkhorn (+1.7%), and Synthesizer (+1.2)%.

The results in Tables 3 and 4 demonstrate some general trends. First, for some attention types, the original 6-8 model aspect ratio never shows optimal performance, *e.g.* the original Transformer model. Second, the nature of the task seems to affect the impact of going wider. Listops saw significant increases compared to the others, perhaps this is due to hierarchical reasoning benefiting from more attention heads. Overall, we would like to highlight the following observations from our results:

- *Wide Transformer networks typically offer equal or sometimes greater accuracy on a range of classification tasks with different sequence lengths.*

- Increases in accuracy are task dependant. Listops had an average accuracy increase of 1.5% when going wide, whereas all the others had changes of < 0.5%.

- The attention mechanism has some effect on whether wider or deeper is better with Longformer and Sinkhorn being more sensitive to the model aspect ratio. We provide possible explanations for these outliers in Section 5.4.

## 5 DISCUSSION

### 5.1 PERFORMANCE & MODEL SIZE

As seen in Section 4, *wide Transformer networks typically offer equal or greater accuracy* on a range of classification tasks with different sequence lengths. The impact of wider and shallower networks on accuracy is slightly influenced by the attention type, with some having more significant changes in accuracy than others. There was no significant difference between convergence time during training for wide and deep models. Each attention mechanism on each task achieved its best validation accuracy after roughly the same number of training steps for all model aspect ratios.

Whilst the total number of parameters involved in the attention layers remains constant amongst different aspect ratios, the overall number of parameters decreases. This is due to the feed-forward network (FFN) part of the Transformer layer remaining unaltered as we change the models width. The models with fewer layers have fewer FFN blocks, and so fewer parameters.

The deepest IMDb byte level classification and Listops models are typically 230MiB, with the widest models typically being 110MiB, only 48% of the size. On token level classification and document matching the sizes are closer. Averaged across all tasks and attention mechanisms, the widest models are 71% the size of the deepest models. A full table of model sizes for the deepest and widest models across all tasks and attention types is given in Table 8, in Appendix B.

### 5.2 LATENCY

As the number of layers in a model decrease, so does the number of dependencies in the computation graph. Because of this a forward pass through the model will have lower latency, though overall throughput may remain unchanged. For systems which require very low latency, such as real time processing in autonomous driving (Talpes et al., 2020), this feature could make a wide single layer model more desirable than an equally accurate deep one.

We measure inference latency for the deepest and widest models for each task and attention type on a CPU with a single input, and on a GPU with a batch input. Experimental details are given in Appendix C along with raw numbers in Tables 9 and 10. We find that on average the widest models are $3.1\times$ faster on a CPU and $1.9\times$ faster on a GPU than the deepest models. The speed-up is consistent across all tasks and attention types.

### 5.3 INTERPRETABILITY

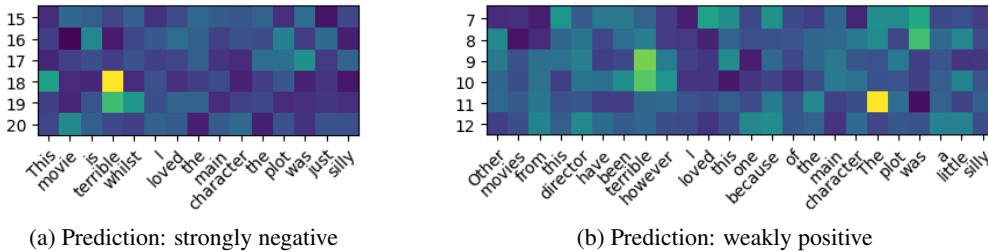

(a) Prediction: strongly negative       (b) Prediction: weakly positive

Figure 2: The attention weights across a selection of heads with predicted classification for the single layer IMDb token level text classification Transformer model on two unseen and similar examples.

Interpretability is increasingly important and a very active area of machine learning research (Gilpin et al., 2018; Linardatos et al., 2020), especially when it comes to fairness. By having more easily inspect-able models, we can see the reasons for a given classification. For example, was a decision based on the mention of a protected characteristic, such as race or gender?

In a Transformer-based architecture, the attention heads in a layer can be inspected during inference to see what connections between input features that head found important. For deep networks, many layers means it can often become unclear what the final output was actually based on (Rigotti et al., 2021). For a single layer wide network, interpretability is far easier as only one layer needs to be inspected. Thus what was considered important for the final output is much clearer.

In Figure 2, we can see the attention weights across some of the heads of the widest token level text classification Transformer model, as well as the predicted class for two different example inputs that have been designed to be similar. Figure 2 includes a subset of the total number of heads, for all 48 heads see Figure 3 in Appendix D.

We can see the review on the left has been confidently assigned as a negative review, and from the attention weights we can see that this is due to the model recognising the relevance of the word "terrible". The review on the right has been less confidently assigned as positive. From the attention weights we can see it has recognised words such as "terrible", but the largest weight is on "The". This explains why the model might not be confident in its positive prediction because it hasn't realised the importance of the word "loved".

## 5.4 Theoretical Explanations of Outliers

Most attention mechanisms usually have up to a 0.5% increase in accuracy when going wider with two notable exceptions. Longformer (Beltagy et al., 2020) typically performs significantly better when deep, and Sinkhorn (Tay et al., 2020a) typically performs significantly better when wide.

For Longformer we only use sliding window attention, with a width of 512. This means, particularly for the longer tasks, each input feature can only have attention computed between it and its neighbours. For deeper models, features can propagate and so this limitation is reduced. However, single layer models suffers a performance penalty.

Sinkhorn works similarly to local attention, where the input sequence is divided into blocks. Unlike local attention, which computes attention within these blocks, Sinkhorn sorts them and computes attention between the original block and the newly sorted block. This sorting mechanism is learnable per head., thus each head can learn a different sorting strategy. For lots of heads this maximises the overall chance of important long range connections within the input sequence being attended to.

## 5.5 Vision Transformer

There is a growing interest in applying Transformers to computer vision (Wang et al., 2021; 2022; Liu et al., 2021; Dosovitskiy et al., 2021). We tested the Pyramid Vision Transformer model (PVT-V2-B1) (Wang et al., 2022) and its wider variants on the CIFAR10 dataset (Krizhevsky et al., 2014). The PVT-V2-B1 model has four stages with each stage containing two attention layers. We then replace the two attention layers at each stage with a single wide attention layer. The detailed architecture of the PVT-V2-B1 model and its wider variants are shown in Table 5. The first wider variant (Wide) matches the total number of heads to the original PVT-V2-B1 model, whereas the later variant (Wide-V2) contains a larger embedding size and more heads. This is a closer match to the original model in terms of the model size.

Table 5: Performance of the original Pyramid Vision Transformer (PVT) and its wider alternatives on the CIFAR10 dataset. The original model (PVTV2-B1) has 4 stages, each stage contains two attention layers. $((1, 1), (2, 2), (5, 5), (8, 8))$ describes the original model, for instance, the first block contains two layers with a single head each, represented as $(1, 1)$.

| Name | Configuration | Accuracy | Parameters |
|---|---|---|---|
| Baseline | $((1, 1), (2, 2), (5, 5), (8, 8))$ | $95.59 \pm 0.99$ | 13.5M |
| Wide | $((2), (4), (10), (16))$ | $94.54 \pm 0.31$ | 7.7M |
| Wide-V2 | $((4), (8), (20), (32))$ | $94.94 \pm 0.20$ | 12.6M |

Table 5 illustrates that the wider variants do not outperform the original PVT-V2-B1 model. Intuitively, spatial features play an important role in vision tasks. The average pooling layer that comes before each attention layer is a crucial component of the PVT model. With only a single wide layer, this pooling layer is not capturing as much spatial information as before. The usage of a single wide attention layer in vision Transformers is constrained by the fact that the majority of these vision Transformers still use pooling or convolution layers before the attention.

Table 6: Test accuracy for each task for wide and deep mixed attention models alongside the averages of all homogeneous models, and the best performing homogeneous model for that task.

| Task | Deepest | | | Widest | | |
|------|---------|---------|-------|---------|---------|-------|
| | Hom. Avg | Hom. Best | Mixed | Hom. Avg | Hom. Best | Mixed |
| IMDb Token Level | 84.8 | 87.0 | 87.3 | 84.7 | **88.0** | 86.8 |
| IMDb Byte Level | 61.3 | 64.5 | 60.1 | 60.7 | **64.8** | 63.0 |
| Listops | 34.2 | 37.1 | 37.1 | 35.7 | 37.7 | **38.0** |
| Document Matching | 63.9 | 71.1 | - | 64.0 | **72.3** | 66.9 |

## 5.6 MIXED ATTENTION

With wider attention layers, the advantages of mixing attention methods becomes more viable. We test a uniform mixture of attentions in both wide and deep variants to see how these models perform. For each layer we use an equal mix of the following attention mechanisms: BigBird, Linear Transformer, Linformer, Local, Longformer, Performer, Sparse Transformer, and Synthesizer.

We omit Sinkhorn since it does not use the [CLS] token for classification like the other attention methods. We also omit vanilla attention so that we have a total of 8 mechanisms, and our overall attention operation is efficient (sub-quadratic time and space complexities).

We initialise a separate multiheaded attention block for each mechanism and average all of their outputs when going back to the sequence features. The number of attention heads in each block is scaled such that the total number equals that of the homogeneous model. For example in the 6 layers, 8 heads configuration, each of our attention blocks has a single head. In the 1 layer 48 heads configuration, each has 6 heads.

Results are given in Table 6. Deep mixed attention on matching is not tested as there are not enough heads to include every attention mechanism. We also include in this table repeats of the averages and bests for each task across all attention mechanisms. From the table we can see that even with mixed attention, the widest single layer models typically perform better (IMDb byte level, Listops) or only marginally worse (IMDb token level) than the deep Transformer models. Whilst beating the averages, all the best mixed models except Listops are outperformed by one of the homogeneous attention models in its widest configuration. The widest mixed model on Listops however points towards mixed attention having possible advantages over homogeneous attention for certain tasks.

## 6 CONCLUSION

When fixing the size of the attention computation, in general wider Tranformers can sometimes outperform, or else match, their deeper counterparts on NLP based tasks. Averaged across all tasks and attention mechanisms, Wide models achieved $+0.3\%$ accuracy compared to Deep models. On most tasks the performance of Wide and Deep was similar, with Listops being the exception ($+1.5\%$). On average going wide works well for almost all attention mechanisms. Sinkhorn benefits the most from being wide, whereas Longformer loses performance.

Furthermore, wider networks are much smaller, only 71% the size of our deepest models on average. They also have additional advantages in interpretability and inference latency: $3.1\times$ faster on CPU and $1.9\times$ faster on GPU for IMDb byte level text classification. On image based tasks however, deeper Tranformers are still superior due to pooling layers being able to capture more spatial information.

Whilst our study considers the effects of attention width across many attention mechanisms and various tasks, it remains to be seen whether these results hold for much larger models on much larger datasets, for example masked langauage modelling. We therefore put forward wider and shallower models as a *viable and desirable alternative* for small models on NLP tasks, and as an *important area of research* for domains beyond this.

## 7 REPRODUCIBILITY STATEMENT

Section 3.1 along with Figure 1 show how we are altering the standard Transformer architecture. The specific tasks we trained our Transformers on are given in Section 3.2 with details on the hyper-parameters used for training in Appendix A. The main body of our code can be found at GIT-LINK.

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

## A  EXPERIMENTAL TASK DETAILS

Table 7: Task specific hyperparameters, see introduction for definitions.

| Task | Training Steps | Warmup | Seq. Length | $E$ | $A$ | $M$ |
|------|---------------|--------|-------------|-----|-----|-----|
| IMDb Token Level | $30k$ | $8k$ | $1k$ | 512 | 64 | 2048 |
| IMDb Byte Level | $30k$ | $8k$ | $1k$ | 512 | 64 | 2048 |
| Listops | $10k$ | $1k$ | $2k$ | 512 | 64 | 2048 |
| Document Matching | $10k$ | $8k$ | $4k$ | 128 | 32 | 512 |

In all tasks we tested on, we used an Adam optimiser with linear warmup and square root decay. We used a base learning rate of 0.05, $\beta_1 = 0.9$, and $\beta_2 = 0.98$ and a batch size of 32 for every task. Attention kernel and MLP weights were initialised via uniform xavier. Biases, input embeddings, and positional embeddings were initialised via Gaussian distributions with variances of $10^{-6}$, 1, and 0.02 respectively. All attention mechanisms, except Sinkhorn, used the [CLS] token for classification. Sinkhorn used mean classifier pooling. Task specific hyperparameters are given in table 7. In all Transformer models the feed-forward network (FFN) has a single hidden dimension. Before testing we take the model which had the best validation accuracy during training, the long train times are to ensure convergence.

## B  MODEL SIZE

Table 8: Model sizes for all tasks and attentions for deepest and widest models (MiB).

| Attention Type | IMDb Token | | IMDb Byte | | Listops | | Doc Matching | | Average | |
|----------------|------|------|------|------|------|------|------|------|------|------|
| | Deep | Wide | Deep | Wide | Deep | Wide | Deep | Wide | Deep | Wide |
| BigBird | 779 | 659 | 230 | 110 | 229 | 109 | 13 | 8.0 | 313 | 222 |
| Linear | 779 | 659 | 230 | 110 | 229 | 109 | 13 | 8.0 | 313 | 222 |
| Linformer | 780 | 659 | 232 | 110 | 231 | 109 | 16 | 8.7 | 315 | 222 |
| Local | 779 | 659 | 230 | 110 | 229 | 109 | 13 | 8.0 | 313 | 222 |
| Longformer | 833 | 713 | 284 | 164 | 283 | 163 | 15 | 11 | 354 | 263 |
| Performer | 779 | 659 | 230 | 110 | 229 | 109 | 13 | 8.0 | 313 | 222 |
| Sinkhorn | 767 | 647 | 218 | 98 | 217 | 97 | 13 | 8.0 | 304 | 213 |
| Sparse | 779 | 659 | 230 | 110 | 229 | 109 | 13 | 8.0 | 313 | 222 |
| Synthesizer | 761 | 641 | 230 | 109 | 300 | 179 | 60 | 55 | 338 | 246 |
| Transformer | 779 | 659 | 230 | 110 | 229 | 109 | 13 | 8.0 | **313** | **222** |
| Average | 782 | 661 | 234 | 114 | 241 | 120 | 18 | 13.1 | **319** | **227** |

Table 8 shows the size of the deepest and widest models for every task and attention combination in mebibytes.

## C  LATENCY

To test latency we input a random full length input into the model 100 times and take the average of the time taken. The latency times for single input on CPU are given in Table 9, the hardware used is $2\times$ AMD EPYC 7763 64-Core Processor 1.8GHz. GPU times are given in Table 10, for this we input a batch of 32 inputs. We use a single Nvidia A100-SXM-80GB GPU.

## D  FULL ATTENTION SCORES

Figure 3 shows the attention scores for every head for our example in Section 5.3.

Table 9: Latency for all tasks and attentions for deepest and widest models on a CPU (ms).

| Attention Type | IMDb Token Deep | IMDb Token Wide | IMDb Byte Deep | IMDb Byte Wide | Listops Deep | Listops Wide | Doc Matching Deep | Doc Matching Wide | Average Deep | Average Wide |
|---|---|---|---|---|---|---|---|---|---|---|
| BigBird | 116 | 46 | 106 | 24 | 103 | 30 | 125 | 33 | 113 | 33 |
| Linear | 54 | 33 | 33 | 11 | 33 | 11 | 26 | 10 | 37 | 16 |
| Linformer | 60 | 34 | 39 | 12 | 39 | 12 | 34 | 12 | 43 | 18 |
| Local | 59 | 34 | 34 | 11 | 33 | 12 | 27 | 11 | 38 | 17 |
| Longformer | 70 | 37 | 62 | 17 | 149 | 37 | 939 | 263 | 305 | 89 |
| Performer | 53 | 33 | 32 | 11 | 55 | 35 | 28 | 9 | 42 | 22 |
| Sinkhorn | 83 | 39 | 59 | 16 | 59 | 17 | 58 | 21 | 65 | 23 |
| Sparse | 56 | 40 | 38 | 14 | 47 | 27 | 88 | 66 | 57 | 37 |
| Synthesizer | 257 | 237 | 866 | 11 | 130 | 20 | 45 | 23 | 324 | 73 |
| Transformer | 60 | 34 | 40 | 13 | 41 | 16 | 56 | 36 | **49** | **25** |
| Average | 87 | 57 | 131 | 14 | 69 | 22 | 140 | 48 | **107** | **35** |

Table 10: Latency for all tasks and attentions for deepest and widest models on a GPU (ms). Averages across attention ignore Synthesizer due to its *extreme* results.

| Attention Type | IMDb Token Deep | IMDb Token Wide | IMDb Byte Deep | IMDb Byte Wide | Listops Deep | Listops Wide | Doc Matching Deep | Doc Matching Wide | Average Deep | Average Wide |
|---|---|---|---|---|---|---|---|---|---|---|
| BigBird | 154 | 63 | 154 | 59 | 210 | 128 | 232 | 115 | 188 | 91 |
| Linear | 52 | 21 | 63 | 37 | 76 | 32 | 52 | 22 | 61 | 28 |
| Linformer | 58 | 21 | 73 | 38 | 81 | 32 | 58 | 23 | 68 | 29 |
| Local | 58 | 26 | 68 | 41 | 86 | 40 | 63 | 34 | 69 | 35 |
| Longformer | 115 | 57 | 83 | 48 | 340 | 189 | 1202 | 540 | 435 | 209 |
| Performer | 47 | 17 | 60 | 35 | 95 | 51 | 49 | 19 | 63 | 31 |
| Sinkhorn | 92 | 33 | 112 | 50 | 121 | 49 | 106 | 52 | 108 | 46 |
| Sparse | 92 | 55 | 76 | 46 | 229 | 175 | 351 | 346 | 187 | 156 |
| Synthesizer | *25164* | 34 | *5927* | *6001* | 325 | 199 | 338 | 293 | 7939 | 1632 |
| Transformer | 88 | 50 | 75 | 47 | 208 | 170 | 289 | 289 | **165** | **139** |
| Average | 76 | 34 | 76 | 40 | 145 | 87 | 240 | 144 | **144** | **76** |

# E   SMALL MODEL ABLATION STUDY

In order to verify the validity of going wide **and** going shallow, we run 'small' vanilla attention Transformer models on all 4 tasks. These models use the same number of heads per layer as the deepest models but have only 1 layer. Therefore 8 heads on both IMDb tasks and Listops, and 4 heads on the document matching task. These results are given in table 11 along with repeats of the averages from the deepest and widest Transformer models. As we can see the small model always performs worse than at least the widest one, though surprisingly beating the deep model on the Listops task. On the IMDb and document matching tasks it consistently performs 1.5%-3% worse than the others, as expected.

Table 11: Test accuracy for each task for the small vanilla Transformer model compared to its fully wide and fully deep counterparts.

| Task | Deepest | Widest | Small |
|---|---|---|---|
| IMDb Token Level | 85.8 | 85.5 | 84.0 |
| IMDb Byte Level | 62.6 | 62.4 | 59.6 |
| Listops | 35.7 | 37.2 | 36.9 |
| Document Matching | 64.0 | 64.4 | 62.4 |

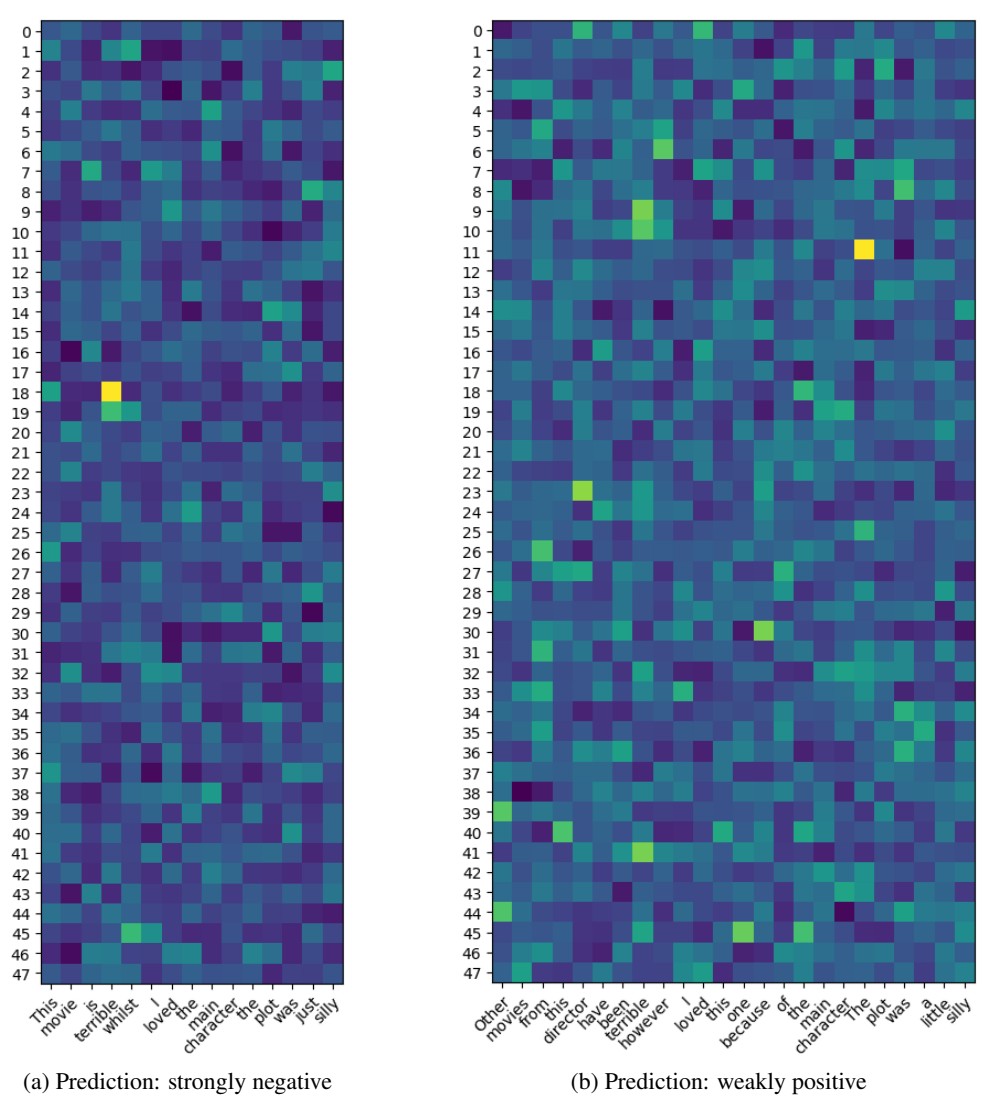

(a) Prediction: strongly negative          (b) Prediction: weakly positive

Figure 3: The attention weights across all 48 heads with predicted classification for the single layer IMDb token level text classification Transformer model on two unseen and similar examples.

