# OpenReview forum: "Wide Attention is the Way Forward for Transformers"
_ICLR.cc/2023/Conference — Submitted to ICLR 2023_

### Official Review · Reviewer_Kk5h · 2022-10-25

**Confidence:** 4
**Correctness:** 3
**Technical Novelty And Significance:** 2
**Empirical Novelty And Significance:** 2
**Recommendation:** 5

**Clarity, Quality, Novelty And Reproducibility:**

The paper is clear and the experiments appear well done.

It should easily be reproducible based on description alone.

The main thrust of the paper (using more heads) is not novel, but using mixed attention is an interesting idea.

**Strength And Weaknesses:**

The paper convincingly lays outs the advantages of their approach. Better classification performance, better latency, and interpretability. The tasks they used are standard benchmarks. Mixed attention is also interesting.

Weaknesses:
- They have restricted themselves to classification tasks. What about language modeling or translation for example?
- They haven't convincingly explained away why models like PaLM would choose to scale to 118 layers and GPT-3 has 96 layers.

**Summary Of The Paper:**

The paper proposes modifying the standard transformer recipe by using more attention heads instead of using layers. Their model has less parameters overall since they keep the feed forward network constant. Despite that, their model demonstrates better performance on a wide range of classification tasks compared to deeper counterparts.

**Summary Of The Review:**

The paper presents an intriguing idea that is worth exploring further, but ultimately I do not feel the that it is novel enough without the further exploration.

---

### Official Review · Reviewer_RFvg · 2022-10-25

**Confidence:** 4
**Correctness:** 3
**Technical Novelty And Significance:** 1
**Empirical Novelty And Significance:** 2
**Recommendation:** 5

**Clarity, Quality, Novelty And Reproducibility:**

The paper is clear and seems mostly reproducible. The direction of exploring shallow transformers is somewhat novel, but it doesn't seem that the authors have been able to show their usefulness for realistic applications, for example with pre-training or for distillation.

**Strength And Weaknesses:**

Strengths:
  * The contains a pretty thorough and principled comparison between deep and wide transformers.
  * The paper does suggest that shallow transformers might be useful as distilled production models, for example if run on CPU. However they don't have experiments in this direction.

Weakness:
  * The study only considers the case of no pre-training. This makes their exploration much easier and less expensive, but their conclusions might not hold at the scales that are currently widely used. In fact the paper cites Xue et al (2022), saying that they come to the conclusion that deep transformers are better than wide in pre-training, although this is for images.
  * The datasets used are only for classification. It would be very nice to see some generation or translation results as well.
  * The interpretability claim is anecdotal and it is not compared against strong baselines for model interpretability in deep models.


**Summary Of The Paper:**

This paper argues that for transformers going deeper isn’t always better, and they show this by proving that wide single layer transformers sometimes outperform deeper transformers in 4 NLP classification tasks, with no pre-training.

The paper reports results on an interesting dimension: for different numbers of attention heads per layer, keeping the total number of heads and other hyper-parameters constant. They run 9 different transformer variations (e.g. BigBird, Longformer, ...), and use different input lengths between 500 and 4000.

The result is that one-layer transformers seem to perform the best on avg by about 0.3%. One-layer transformers are also smaller in terms of number of parameters 1.4x and faster to run (on CPU 3.1x times faster, on GPU 1,9x).

The authors also claim that one-layer transformers are also more interpretable showing some examples, but this is a difficult claim to make objectively.


**Summary Of The Review:**

The lack of results for pretraining really weakens the paper, since virtually all current interesting results in NLP are based on massive pretraining.

Given that the authors claim improvements in speed and footprint, they could try making this contribution stronger by comparing against prior work optimizing transformers for efficiency.

---

### Official Review · Reviewer_xaMf · 2022-10-25

**Confidence:** 4
**Correctness:** 1
**Technical Novelty And Significance:** 1
**Empirical Novelty And Significance:** 2
**Recommendation:** 3

**Clarity, Quality, Novelty And Reproducibility:**

- The paper is easy to follow
- Technical novelty is not its strength. There are flaws in its claims.
- I am not able to determine its reproducibility based on the experimental detail the paper provides.

**Strength And Weaknesses:**

Strength:
- Straightforward idea.
- Writing is reasonably clear.
- It’s great that the paper considers CPUs in addition to GPUs in efficiency comparisons.

Weaknesses:
- The paper overclaims: it opens with a general claim that going wide is better than going deep on a “variety of NLP tasks,” while the experiment is on three toy text encoding datasets.
- The paper fails to acknowledge existing findings. The wider vs. deeper debate has been there for perhaps two decades, and recent results suggest that going deeper leads to better generalization. The paper argues for the opposite without discussing any existing work.

Further details and comments:
- The wider vs. deeper debate has been there for a while. To convince the community that going wider is better, the paper needs to do a much better job. I suggest starting by covering a diverse set of tasks, e.g., language modeling, text generation, QA; and then considering larger-scale LM/MLM pretraining. Having a more comprehensive evaluation of the transformer model is more relevant than covering all the X-formers on toy datasets.
- Given the existing work, my prior is that it is challenging (if at all possible) to match deep transformers’ performance with wide and shallow models on most tasks. If this is the case, I’d encourage the authors to explore the efficiency direction: can wider models achieve a better tradeoff between accuracy and efficiency?
- I don’t get the 5.3 arguments on interpretability. Is there anything you can do with a wide transformer that cannot be done with a deep one?


**Summary Of The Paper:**

This paper investigates deep vs. wide (in terms of the number of attention heads) models with a variety of recent transformer variants. It argues that, compared to their deep and thin counterparts, shallow and wide architectures (1) achieve the same or better performance, (2) has fewer parameters, and (3) run faster. The experiments on four text classification datasets provide evidence supporting the claim, while those on image classification yield negative results.




**Summary Of The Review:**

The paper has serious flaws in its claims and is not ready for publication.

---

> ### Comment · Reviewer_xaMf · 2022-11-18
> **After author response**
>
> After reading the revision and the authors' response I decide to keep my score unchanged.

---

### Official Review · Reviewer_MwhQ · 2022-10-26

**Confidence:** 4
**Clarity, Quality, Novelty And Reproducibility:** This work is clear, well-written, and…
**Correctness:** 3
**Technical Novelty And Significance:** 2
**Empirical Novelty And Significance:** 2
**Recommendation:** 3

**Strength And Weaknesses:**

Strengths:
1. Surprising results that a single-layer transformer works as good as multi-layer transformers on sequence classification problems including listops which was constructed to test reasoning and presumably requires deep models.

Weaknesses:
1. I think the results do not support the title that wide attention is the way forward for transformers, since experiments are run on classification type problems, and it's not clear whether a single-layer transformer works on other types of problems such as text generation.
2. While having fewer parameters is considered a plus, it is also possible that single-layer transformers overfit less due to having fewer parameters. What if you use really large datasets and more complex tasks such that model capacity becomes crucial? To convince readers that it's not the case, an experiment where the number of parameters of deep transformers is reduced to the same as single-layer transformers would be nice.
3. In my opinion a fundamental limitation of single-layer models is that there is a single feedforward layer, which might lack the capacity to learn more complex tasks. A counter argument might be that we can increase the hidden dimension of the feedforward layer to increase capacity, but that comes back to general neural architecture search, and a really convincing experiment would be to run these experiments on more complex types of problems such as sequence generation.
4. Regarding the interpretability argument, I'm not fully convinced it's more interpretable looking at Figure 3. It seems that we can pick one or two attention heads out of 48 heads that are interpretable, but I think we can do something similar in deep transformers.

Typos:
1. page 6: eg. the original

**Summary Of The Paper:**

This work empirically studies decreasing the number of layers in transformers while increasing the number of attention heads. On four classification problems and across many attention variants, this work found that a transformer with a single layer but more attention heads (such that the total number of attention heads remains constant) has on par or better accuracy than multi-layer transformers. This paper also argues that the attention matrices in a single-layer transformer are easier to interpret. Besides, results on image classification show that deep transformers work better there.

**Summary Of The Review:**

I'm not recommending the acceptance of this paper because to me this work only shows that single-layer transformers work well for simple classification problems. I would be convinced the other way if the authors can show similar trends on more complex tasks such as sequence generation.

---

### Decision · Program_Chairs · 2023-01-20

**Decision:**

Reject

**Justification For Why Not Higher Score:**

All the reviewers are in consensus that the results do not warrant the strong claim in the title of the paper. Unfortunately the authors didn't provide any response to reviewer concerns. The experiments are only carried on simple classification tasks and reported numbers are close to those obtained by bag of words models like logistic regression on IMDB (86%). It would be more convincing if wide single layer transformer was shown to match state of the art deep transformer accuracy (> 95%). Furthermore, evaluation on more complex tasks like translation, question answering, etc. would be essential to support the claim.

**Justification For Why Not Lower Score:**

N/A

**Metareview: Summary, Strengths And Weaknesses:**

The paper attempts to improve the usage of the popular transformer architecture and make them your efficient. In this regards, the authors empirically studies a different configuration of transformer: make them wider and shallower instead of deeper. In particular, decrease the number of layers while increasing the number of attention heads per layer while keeping their product constant. On four simple classification tasks, this work found that a single layer but more attention heads has on par or better accuracy than multi-layer transformers. However, all the reviewers are in consensus that the results do not warrant the strong claim in the title of the paper. Unfortunately the authors didn't provide any response to reviewer concerns. The experiments are only carried on simple classification tasks and reported numbers are close to those obtained by bag of words models like logistic regression on IMDB (86%). It would be more convincing if wide single layer transformer was shown to match state of the art deep transformer accuracy (> 95%). Furthermore, evaluation on more complex tasks like translation, question answering, etc. would be essential to support the claim that "wide attention is the way forward for transformers".

**Summary Of Ac-Reviewer Meeting:**

N/A